# Personalized Multi-tier Federated Learning

**Sourasekhar Banerjee**
Dept. of Computing Science
Umeå University
Sweden
sourasb@cs.umu.se

**Alp Yurtsever**
Dept. of Mathematics
and Mathematical Statistics
Umeå University
Sweden
alp.yurtsever@umu.se

**Monowar Bhuyan**
Dept of Computing Science
Umeå University
Sweden
monowar@cs.umu.se

## Abstract

The challenge of personalized federated learning (pFL) is to capture the heterogeneity properties of data with inexpensive communications and achieving customized performance for devices. To address that challenge, we introduced personalized multi-tier federated learning using Moreau envelopes (pFedMT) when there are known cluster structures within devices. Moreau envelopes are used as the devices' and teams' regularized loss functions. Empirically, we verify that the personalized model performs better than vanilla FedAvg, per-FedAvg, and pFedMe. pFedMT achieves 98.30% and 99.71% accuracy on MNIST dataset under convex and non-convex settings, respectively.

## 1 introduction

Federated learning (FL) is a distributed on-device learning framework that employs the heterogeneous data privately available at the edge for learning. In traditional machine learning, edge devices are supposed to send data to the global server for training. However, federated learning relaxes this restriction by enabling training each model on the end devices and aggregate them on the global server. Classical federated learning has an objective to compute a single global model from the device's private data in a communication-efficient and privacy-preserving manner [1]. Virtually all methods in FL follow the following basic steps: (1) The global server sends the initial global model to all participating devices. (2) A subset of devices is selected to compute local models. (3) Each selected device learns a local model with their private data. (4) The global server collects and aggregates all the local models, produces the global model, and sends it back to all participating devices. The steps 2 to 4 continues until the desired accuracy is achieved. FL suffers from data and system heterogeneity among devices with size and distribution of local data as well as storage and computation capacity, respectively. These challenges adversely impact the model's convergence and learning efficiency. Data heterogeneity means the data is disseminated across devices in a non-independent and identical manner (non-IID). That raises a problem in generalizing the global model for each device. Unlike conventional FL, personalized FL aims to learn device-specific models along with a global model to reduce the generalization error and achieve customize learning performance [2].

In FL, it is typically assumed that devices can be grouped into a single cluster [1, 3], and there will be a single global server to which the devices communicate. Multi-tier FL differs from the vanilla

Workshop on Federated Learning: Recent Advances and New Challenges, in Conjunction with NeurIPS 2022 (FL-NeurIPS'22). This workshop does not have official proceedings and this paper is non-archival.

FL architecture. Here, devices have no direct communication with the global server. Another layer called "team" is present between the global server and devices that have direct contact with the global server. The idea of multi-tier FL is similar to the Cloud-Edge model. The cloud acts as a "global server", responsible for producing a global model. Edge servers in different geographical regions acts as "team server" and connect with the cloud and end devices. End devices compute the local model. Multi-tier framework is more realistic to solve the real world federated learning problems that includes Cloud-Edge architecture [4].

A non-smooth function can be made smooth using the Moreau envelope. Motivated by the excellent performance of Moreau envelope [5] in personalized FL [2], we propose multi-tier federated learning where devices simultaneously minimize the Moreau envelopes of the device loss function and build the global model as in the classical FL. In our proposed architecture, we leverage the hierarchical architecture that has three levels, such as global server, team, and device, and simultaneously learn (1) a personalized model for each device, (2) a personalized model for each team, and (3) a global model. Geometrically, under this multi-tier system, the global model can be seen as a "central point" where all teams agree. In contrast, personalized models are the points in various directions that devices take in accordance with their data distributions. Our algorithm only allows devices to communicate within their team. Team servers of each team communicates with the global server. Overall objective of pFedMT is to achieve customized on-device model performance as well as improve performance in global models by in-expensive number of global iterations.

**Our key contributions** are as follows: First we formulate the optimization problem for multi-tier personalized FL (pFedMT) by using Moreau envelope as a regularized loss function for both devices and teams.

Second, we designed the algorithm for pFedMT on the basis of multi-tier FL architecture which works for both convex and non-convex settings.

Finally, we empirically evaluated the performance of pFedMT using both benchmark and synthetic datasets in non-iid settings. pFedMT outperforms the vanilla FedAvg [1], Personalized FedAvg [6], and pFedMe [2] for both convex and non-convex settings. From the empirical analysis, we have also shown that pFedMT achieves customize on-device performance with in-expensive communication iterations.

The rest of the paper is organized as follows: Related work is discussed in Section 2. After that, in Section 3 we proposed pFedMT. Experiments and results are given in Section 4.

## 2 Related Work

Our work is relevant to hierarchical, multi-task, and personalized federated learning; therefore, here we provide a brief literature survey on these.

**Hierarchical federated Learning** Most of the recent work in FL assumes a single global parameter server, which may be located either at the edge or in the cloud. The cloud server has more access to data but is not communication efficient and expected high latency, whereas the edge servers are more communication efficient with devices. The hierarchical model proposed in [7] takes advantage of both the cloud and edge in federated learning. In [8], authors introduced the upward and downward divergences in hierarchical FL. Another work in [9] applied personalization in hierarchical FL to reduce the generalization error at each device.

**Multi-task federated learning** MOCHA [3] was one of the earliest works on multi-task learning (MTL) in federated settings that considered issues of high communication cost, stragglers, and fault tolerance for distributed multi-task learning. Multi-task learning helps to handle statistical heterogeneity problems and extended to personalization. In [10], proposed an MTL approach under a mixture of distributions in federated settings. Duan et. al. [11] introduced a group of adaptive techniques that carefully manage task differences while automatically utilizing potential similarities.

**Personalized federated learning** Due to its simplicity and low connection cost, FedAvg [1] has become the preferred method for federated learning. But it suffers from concept-drift when the data is heterogeneous (non-iid), leading to unstable and sluggish convergence. Concept drift is a problem where no single global model works well for all clients. To solve this problem and generate

a better convergence rate, over the past few years, FL has placed much emphasis on personalization. SCAFFFOLD [12] is one of the earliest algorithm that requires fewer communication iterations without affected by data heterogeneity. In [6], a personalized version of model-agnostic meta-learning (MAML) [13] is proposed for federated learning. The main objective of that work is to finding an initial shared model that users may quickly adapt to their local dataset by executing few gradient descent steps using their own data. In [14], authors conducted a systematic learning-theoretic study of personalization and proposed three approaches: user clustering, data interpolation, and model interpolation, that are model-agnostic. pFedMe [2], authors formulate personalized FL by using Moreau envelopes as devices' regularized loss function. In [9], authors proposed a hierarchical personalized FL approach where the users come from multiple known clusters. The algorithm simultaneously learns three models: a global model, a cluster-specific model for each cluster, and a personalized model for every device. The architecture is similar to pFedMT, except that we applied Moreau envelope as a regularizer. Few more recent works on personalized FL are as follows, [15, 16, 17].

The motivation behind pFedMT is two-fold: One, it is reasonable to assume that closer devices contain similar information but data is more heterogeneous for distant devices. By adding personalization both at team and device levels, we aim to capture a more elaborate personalization model that aligns better with the real-world applications. Two, communication with the global server is often the most expensive step in FL, but communications within a team is typically much cheaper. By exploiting the multi-tier architecture and accommodating a large portion of the communication within the teams, pFedMT economizes significantly on the communication iterations with the global server.

## 3  pFedMT

In a multi-tier FL framework (Figure 1), all devices are divided into M teams. Each team has a team server ($TS_i$) and it is connected with $N_i$ devices. Devices belonging to the same team are supposed to come from similar geographical locations; therefore, devices within a team have more similar models than devices belonging to another team. Devices within a particular team only communicate with the respective team server($N_i$). Similarly global server ($GS$) only communicates with the team server for global updates.

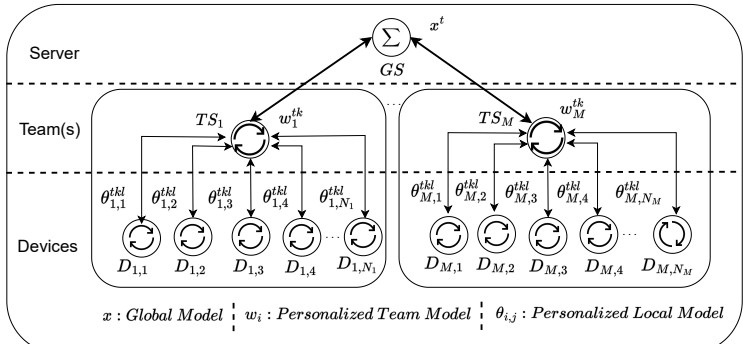

Figure 1: pFedMT: a proposed learning framework

### 3.1  pFedMT formulation

We consider a multi-tier FL setup with $M$ teams, with $N_i$ devices in each team for $i = 1, \ldots, M$. The standard empirical risk minimization for this problem can be written as

$$\underset{x \in \mathbb{R}^d}{\text{minimize}} \quad \frac{1}{M} \sum_{i=1}^{M} \frac{1}{N_i} \sum_{j=1}^{N_i} f_{i,j}(x), \tag{1}$$

where $f_{i,j}(\cdot)$ is the loss function of $j^{th}$ device from $i^{th}$ team. This formulation involves a single decision variable $x$ that all clients are expected to agree on when the model converges. This is unsuited for problems in which the clients' data distributions are non-homogenous because a global

model that works for all devices may be nonexistent. This problem can be addressed effectively by replacing the local objective functions with their Moreau envelopes [2]. We design a multi-tier personalized FL formulation by following this idea.

Let us first define the team-level objective function with device-level personalization:

$$F_i(w_i) := \frac{1}{N_i} \sum_{j=1}^{N_i} \tilde{f}_{i,j}(w_i), \quad \text{where} \quad \tilde{f}_{i,j}(w_i) = \min_{\theta_{i,j} \in \mathbb{R}^d} \left( f_{i,j}(\theta_{i,j}) + \frac{\lambda}{2} \|\theta_{i,j} - w_i\|^2 \right). \quad (2)$$

$\tilde{f}_{i,j}(\cdot)$ is the Moreau envelope of $f_{i,j}(\cdot)$, $w_i$ is the model of $i^{th}$ team, $\theta_{i,j}$ is the personalized model of $j^{th}$ device in $i^{th}$ team, and $\lambda \geq 0$ is a hyperparameter that controls the device-level personalization impact. We use Moreau envelope again on the team-level objective to obtain our global loss function,

$$\phi(x) := \frac{1}{M} \sum_{i=1}^{M} \tilde{F}_i(x) \quad \text{where} \quad \tilde{F}_i(x) = \min_{w_i \in \mathbb{R}^d} \left( F_i(w_i) + \frac{\gamma}{2} \|w_i - x\|^2 \right). \quad (3)$$

Here, $\tilde{F}_i(\cdot)$ is the Moreau envelope of $F_i(\cdot)$, $x$ is the global (server) model, $w_i$ is the personalized model of $i^{th}$ team, and $\gamma \geq 0$ is a hyperparameter that controls the team-level pesonalization.

The goal in pFedMT is to minimize $\phi(x)$. We use the gradient method: Starting from an initial estimate $x^0 \in \mathbb{R}^d$, and for a given step-size $\beta > 0$, for $t = 0, 1, \ldots, T$

$$x^{t+1} = x^t - \beta \nabla \phi(x^t) = x^t - \frac{\beta}{M} \sum_{i=1}^{M} \nabla \tilde{F}_i(x^t) = (1 - \beta\gamma)x^t + \frac{\beta}{M} \sum_{i=1}^{M} \gamma \text{prox}_{F_i/\gamma}(x^t), \quad (4)$$

where the last equality follows from the definition of the gradient of Moreau envelope:

$$\nabla \tilde{F}_i(x^t) = \gamma \big( x^t - \text{prox}_{F_i/\gamma}(x^t) \big). \quad (5)$$

In general, $\text{prox}_{F_i/\gamma}(x^t)$ can be difficult to compute, but we can compute it by considering the Moreau envelope as a subproblem,

$$\underset{w_i \in \mathbb{R}^d}{\text{minimize}} \quad F_i(w_i) + \frac{\gamma}{2} \|w_i - x^t\|^2. \quad (6)$$

At every global round $t$, each team approximates the solution of Problem 6 by using the gradient method. Starting from $w_i^{t,0} = x^t$, and for given step-size $\eta_i > 0$, we compute for $k = 0, 1, \ldots, K$,

$$\begin{aligned}
w_i^{t,k+1} &= w_i^{t,k} - \eta_i \nabla F_i(w_i^{t,k}) - \eta_i \gamma(w_i^{t,k} - x^t) \\
&= w_i^{t,k} - \frac{\eta_i}{N_i} \sum_{j=1}^{N_i} \nabla \tilde{f}_{i,j}(w_i^{t,k}) - \eta_i \gamma(w_i^{t,k} - x^t) \\
&= (1 - \eta_i\lambda - \eta_i\gamma)w_i^{t,k} + \eta_i\gamma x^t + \frac{\lambda\eta_i}{N_i} \sum_{j=1}^{N_i} \text{prox}_{f_{i,j}/\lambda}(w_i^{t,k}),
\end{aligned} \quad (7)$$

where the last equality follows from the definition of the gradient of Moreau envelope:

$$\nabla \tilde{f}_{i,j}(w_i^{t,k}) = \lambda \big( w_i^{t,k} - \text{prox}_{f_{i,j}/\lambda}(w_i^{t,k}) \big). \quad (8)$$

Once again, $\text{prox}_{f_{i,j}/\lambda}(w_i^{t,k})$ can be difficult to compute. Therefore we handle this operation as a subproblem solved at the device-level:

$$\underset{\theta_{i,j} \in \mathbb{R}^d}{\text{minimize}} \quad f_{i,j}(\theta_{i,j}) + \frac{\lambda}{2} \|\theta_{i,j} - w_i^{t,k}\|^2. \quad (9)$$

In every team-level update, each device approximate $\text{prox}_{f_{i,j}/\lambda}(w_i^{t,k})$ by applying gradient method to problem 9. Starting from $\theta_{i,j}^{t,k,0} = w_i^{t,k}$, and using a step-size $\alpha > 0$, we compute for $l = 0, 1, \ldots, L$,

$$\theta_{i,j}^{t,k,l+1} = \theta_{i,j}^{t,k,l} - \alpha \nabla f_{i,j}(\theta_{i,j}^{t,k,l}) - \alpha\lambda(\theta_{i,j}^{t,k,l} - w_i^{t,k}). \quad (10)$$

This framework gives us a triple-loop mechanism for personalized multi-tier FL, described below.

## 3.2 Algorithm

We propose an algorithm for personalized multi-tier federated learning using Moreau envelope (pFedMT). Global server initializes the global model ($x^0$). Every team connected to the global server, copies the global model to their respective team model ($\forall_{i=1}^M w_i^{0,0} = x^0$). Each device within a team copies the initial team model ($w_i^{0,0}$) as their local model($\theta_{i,j}$). Global server initializes total number of global iterations, team iterations, and local iterations as $T, K$, and $L$ respectively.

At each global iteration $t$ (step 1), global server broadcasts global model $x^t$ to every teams (step 2). Similarly, at each team iteration $k$, each team boradcasts $w_i^{t,k}$ to all the devices within the team (step 4). For each local iteration ($l$), each device $D_{i,j}$ solves Equation 10 separately but in parallel to obtain the personalized model $\theta_{i,j}^{t,k,l}$ (steps 5 and 6). The team server ($N_i$) of each team collects the device updates from the respective devices after $L$ local iterations and perform aggregation ($\bar{\theta}_i^{t,k}$) on the device updates (step 8) which is similar to weighted FedAvg [18]. $\delta_j$ is a weighting factor based on the ratio between the device's number of samples and the total number of samples. And also, $\sum_{j=1}^{N_i} \delta_j = 1$. Each $N_i$ produce personalized model ($w_i^{t,k}$) using Equation 7(step 9). Each team broadcast the updated team level model to the devices registered with that team and continue steps 3 to 10 for next $K - 1$ team iterations.

After all teams finished $K$ team iterations, the global server collects team updates ($w_i^{t,K}$) from each team and performs weighted aggregation ($w^t$) on team updates (step 11). $\tau_i$ is a weighting factor that depends on the ratio between the number of samples of the team and the total amount of data. Similar to $\delta_j$, $\sum_{i=1}^M \tau_i = 1$. The global server produces a global update($x^t$) by solving Equation 4 (step 12). The global aggregation and global update both would be similar if value of $\gamma$ and $\beta = 1$. The global server broadcasts the updated global model to teams and continues the steps 1 to 12 for the upcoming $T - 1$ global iterations.

---

**Algorithm 1** pFedMT : Personalized Multi-tier Federated Learning using Moreau Envelope

---

    **Input :** $x^0$
    **Initialize :** $\forall_{i=1}^M w_i^{0,0} = x^0$ , $\forall_{i=1}^M \forall_{j=1}^{N_i} \theta_{i,j} = w_i^{0,0}, T, K, L, \alpha_{i,j}, \beta, \gamma, \lambda, \eta_i$
    **Output :** $\forall_{i=1}^M \forall_{j=1}^{N_i} \theta_{i,j}^{T,K,L}, x^T$

1: **for** $t = 0, 1, \ldots, T$ **do**                                                         ▷ Global iterations
2:     global server sends $x^t$ to the teams.                                 ▷ Global model
3:     **for** $k = 0, 1, \ldots, K$ **do**                                             ▷ Team iterations
4:         Teams send $w^{t,k}$ to the devices.                      ▷ Team-level personalized model
5:         **for** $l = 0, 1, \ldots, L$ **do.**                                      ▷ Local iterations
6:             $\theta_{i,j}^{t,k,l+1} = \theta_{i,j}^{t,k,l} - \alpha_{i,j} \nabla f_{i,j}(\theta_{i,j}^{t,k,l}) - \alpha_{i,j} \lambda (\theta_{i,j}^{t,k,l} - w_i^{t,k})$      ▷ Personalized local models
7:         **end for**
8:         $\bar{\theta}_i^{t,k} = \sum_{j=1}^{N_i} \delta_j \theta_{i,j}^{t,k,L}$                               ▷ Aggregation within a team
9:         $w_i^{t,k+1} = (1 - \eta_i \lambda - \eta_i \gamma) w_i^{t,k} + \eta_i \gamma x^t + \lambda \eta_i \bar{\theta}_i^{t,k}$        ▷ Personalized team update
10:     **end for**
11:     $\bar{w}^t = \sum_{i=1}^M \tau_i w_i^{t,K}$                                  ▷ Global aggregation
12:     $x^{t+1} = (1 - \beta \gamma) x^t + \beta \bar{w}^t$                             ▷ Global update
13: **end for**

---

# 4 Experiments

We study a classification problem to validate pFedMT using both benchmark (MNIST [19]) and synthetic datasets. Both datasets contain ten class labels. We divided MNIST in a non-iid manner into two teams, each containing 10 devices. We adopt the non-iid data generation and distribution procedure from [2] for synthetic data preparation. Similarly, the generated synthetic data is distributed into two teams, each containing 25 devices. We took into account an $l_2$-regularized multinomial logistic regression (MLR) with softmax activation loss function for $\mu$-strongly convex settings. A two-layered deep neural network for synthetic dataset and a convolutional neural network is built for MNIST dataset as non-convex settings. All datasets are split into train and test sets with a ratio of 3:1. For the entire experiments we used the learning rate $\alpha = 0.01$. For pFedMT, we set $\eta = 0.03$, and $\gamma = 0.1$ for convex settings, and $\eta = 0.03$ and $\gamma = 1$ for non-convex settiings. The value of $\lambda, \gamma, \beta$ is given in Table 1 for the respective experiments. We consider the value of $T$ = 100 and 600, $K$ = 30 and $L$ = 20 for pFedMT. The personalized models and global model of pFedMT are represented by

pFedMT(PM) and pFedMT(GM) respectively. Similarly, pFedMe(PM) and pFedMe(GM) denotes personalized and global model of pFedMe, respectively.

## 4.1 Results and Analysis

We analyzed the communication efficiency and convergence of pFedMT with the state-of-the-art (SOTA) models. From Figure 2, we observed convergence of the personalized model pFedMT(PM) and global model pFedMT(GM) are quicker than the personalized pFedMe(PM) and global model pFedMe(GM) of pFedMe, and Per-FedAvg, respectively. pFedMT(PM) converges within fewer than 100 global iterations. The global model needs more iterations to converge.

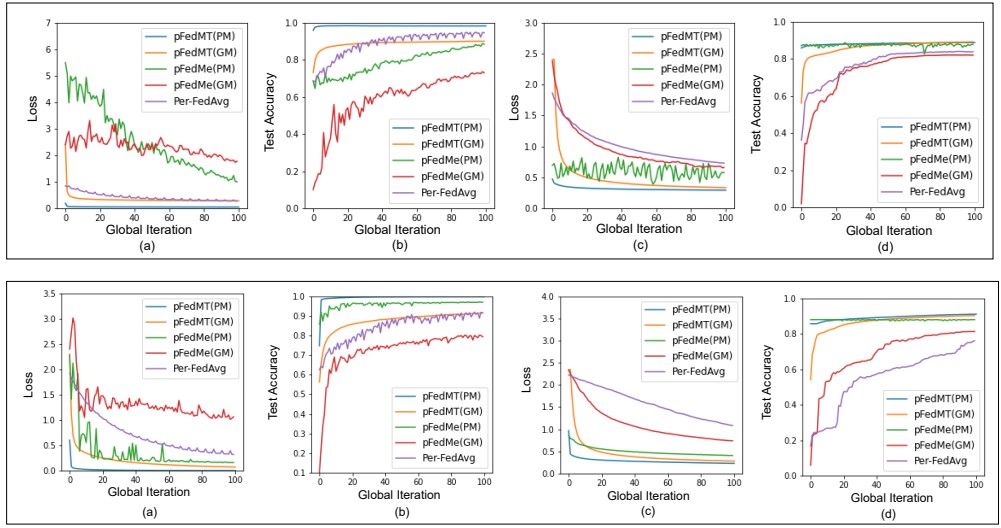

Figure 2: Convergence comparison in convex *[top]* and non-convex *[bottom]* settings using - MNIST (a) Loss (b) Test accuracy - synthetic data (c) Loss, (d) Test accuracy

From Table 1, we observed both convex and non-convex cases. The pFedMT(PM) outperforms the SOTA. pFedMT(GM) is better than the FedAvg on both MNIST and synthetic datasets. For synthetic datasets, pFedMT(GM) performs better than per-FedAvg and pFedMe. In convex settings, pFedMT(PM) reached 98.30% test accuracy within 100 global iterations. The closest one is Per-FedAvg which reaches 96.26% after 600 global iterations. On synthetic datasets after training of 600 global iterations, pFedMT(GM) produces slightly better accuracy than pFedMT(PM). In non-convex settings, the test accuracy of pFedMT(PM) on MNIST datasets reaches 99.71%, which is 0.38% higher than the pFedMe for 600 global iterations. On synthetic datasets using deep neural network (DNN), pFedMT(PM) and pFedMT(GM) produce, 95.42% and 94.05% test accuracy, respectively, after training for 600 iterations. Whereas pFedMe(PM) can produce, 89.53% accuracy on the similar settings. From the results, we can say that empirically pFedMT(PM) outperforms the SOTA.

Table 1: Results on MNIST and synthetic datasets

| | | | | MLR | | | | | | |
|---|---|---|---|---|---|---|---|---|---|---|
| Algorithm | | MNIST | | | | | Synthetic | | | |
| | $\lambda$ | $\gamma$ | $\beta$ | Accuracy(%) (T = 100) | Accuracy(%) (T = 600) | $\lambda$ | $\gamma$ | $\beta$ | Accuracy(%) (T = 100) | Accuracy(%) (T = 600) |
| FedAvg | - | - | - | 84.87 (±0.054) | 89.70 (±0.00) | - | - | - | 84.87(±0.054) | 89.004(± 0.08 ) |
| Per-FedAvg | - | - | - | 94.81(±0.00) | 96.26 (±0.00) | - | - | - | 83.91(±0.15) | 86.84(±0.16) |
| pFedMe(GM) | 15 | - | 0.5 | 75.50(±0.00) | 85.47(±0.00) | 15 | - | 0.5 | 81.93(±0.21) | 85.65(±0.25) |
| pFedMe(PM) | 15 | - | 0.5 | 88.89(±0.001) | 94.53(±0.001) | 15 | - | 0.5 | 88.61(±0.16) | 89.08 (±0.10) |
| **pFedMT(GM)** | 15 | 0.1 | 1.0 | 89.92(±0.001) | 90.22 (±0.00) | 15 | 0.1 | 1.0 | 88.67(±0.79) | **90.54 (±0.02)** |
| **pFedMT(PM)** | 15 | 0.1 | 1.0 | **98.30**(±0.01) | **98.30**(±0.01) | 15 | 0.1 | 1.0 | **89.20**(±0.56) | 89.86 (±0.02) |
| | | | | DNN or CNN | | | | | | |
| FedAvg | - | - | - | 93.17 (±0.02) | 98.79 (±0.03) | - | - | - | 84.53(±0.067) | 89.96(±0.061) |
| Per-FedAvg | - | - | - | 91.845(±0.00) | 96.65(±0.00) | - | - | - | 75.93 (±0.18) | 84.71(±0.20) |
| pFedMe(GM) | 15 | - | 0.4 | 80.12(±0.01) | 86.86(±0.00) | 15.0 | - | 0.5 | 81.23(±0.19) | 84.17 (±0.35) |
| pFedMe(PM) | 15 | - | 0.4 | 97.40(±0.00) | 98.33(±0.001) | 15.0 | - | 0.5 | 87.86(±0.06) | 89.53(±0.01) |
| **pFedMT(GM)** | 3.0 | 1.0 | 1.0 | 93.49(±0.008) | 96.72 (±0.01) | 15.0 | 1.0 | 1.0 | 90.96(±2.01) | 94.05(±0.08) |
| **pFedMT(PM)** | 3.0 | 1.0 | 1.0 | **99.47**(±0.004) | **99.71** (±0.01) | 15.0 | 1.0 | 1.0 | **92.196**(±1.59) | **95.42**(±0.01) |

## 4.2 Conclusions

We proposed pFedMT, a personalized multi-tier federated learning system that consists of a global server, teams, and devices. We applied the Moreau envelope as a regularized loss function on devices and teams. pFedMT produces device-specific personalized models and a global model simultaneously. From the empirical results, we observed that pFedMT is communication efficient and well generalized model across devices. This work is still in its early stage. Therefore, in the near future, we will provide theoretical guarantees of pFedMT, empirical analysis of the effect of hyperparameters $\lambda$ and $\gamma$, and extensive experiments with state-of-the-art models.

## Acknowledgments and Disclosure of Funding

This work was partially supported by the Wallenberg AI, Autonomous Systems and Software Program (WASP) funded by Knut and Alice Wallenberg Foundation. The computations were enabled by the supercomputing resource Berzelius provided by National Supercomputer Centre at Linköping University and the Knut and Alice Wallenberg foundation.

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
