# OpenReview forum: "Personalized Multi-tier Federated Learning"
_NeurIPS.cc/2022/Workshop/Federated_Learning — FL-NeurIPS 2022 Poster_

### Official Review · Reviewer_4gkp · 2022-10-12
**Interesting and Clearly Presented Approach to Clustered-Personalization**

The paper presents pFedMT, multi-tier personalization for federated learning through regularization between global, regional (team), and local (individual) models. The objectives are clearly stated and easy to follow, the intuition behind the idea is clear, and the results are strong. The approach requires the identification of regions/teams for multi-tier learning (i.e., which subset of users should be assigned a shared team?) and otherwise is not particularly novel, but the results are strong and the method is clear.

Strengths:
* The proposed approach is easy to follow and clearly described.
* pFedMT achieves strong performance, both in the global and local models, suggesting that added regularization between similar clusters of users is indeed helpful for personalized federated learning models.
* The paper compares against another personalized federated learning paper (pFedMe) that does not use multi-tier learning, showing the value in their tiered approach.

Weaknesses:
* The discovery or creation of teams is not addressed in this work, though it is a crucial component of successfully deploying pFedMT. No experiments or ablations are performed to examine worst-case performance if teams are poorly constructed.
* The experiments only include up to 25 devices and two teams, so evaluating the impact of more diverse or large-scale populations is not possible from these results.
* Small clarity issues throughout (See below)

Clarity:
* Line 125: t and T are not defined.
* Line 130: k and K are not defined.
* Line 135: l and L are not defined until later in the paper, so this is hard to interpret.
* Figure 2: The colors are not distinct enough for colorblind viewers, making the plot impossible to interpret. The table of results is helpful, but for these plots to be more usable they should have line markers and/or be larger.
* Experiments: How many runs are performed to get standard deviation numbers in the results?

---

### Official Review · Reviewer_Vtm7 · 2022-10-19
**The paper proposes a novel approach for training global and personalized models simultaneously using a multi-tier federated learning algorithm. However, despite having good empirical results, the experimental setup is not convincing enough to justify the conclusions.**

The proposed approach groups similar clients into teams and applies a Moreau envelope as a regularized loss function on those teams. The method trains a global model as well as separate personalized models for each client. The authors show empirical results on the MNIST dataset and show it outperforms baselines.

Strengths:
+Empirical results show good performance.
+The use of multi-tier learning is a novel idea

Weakness:
-The paper only performs analysis on MNIST dataset which is not convincing enough as the paper hinges on the empirical results.
-The paper lacks a theoretical foundation.
-The paper’s experiment section is not compelling enough.

Questions:
- Can the author provide a reference that supports their assumption that “Devices belonging to the same team are supposed to come from similar geographical locations; therefore, devices within a team have more similar models than devices belonging to another team.” How does geographical location relate to the model?
- What is the performance of the clients if they were randomly grouped instead? A sharp difference between the random and proposed approach will help justify its performance.

---

### Decision · Program_Chairs · 2022-10-20

Accept (Poster)